# Goal-Conditioned On-Policy Reinforcement Learning

**Xudong Gong**[1,2,†]    **Dawei Feng**[1,2,†]    **Kele Xu**[1,2,*]    **Bo Ding**[1,2*]    **Huaimin Wang**[1,2]

[1] College of Computer, National University of Defense Technology, Changsha, Hunan, China
[2] State Key Laboratory of Complex & Critical Software Environment, Changsha, Hunan, China

## Abstract

Existing Goal-Conditioned Reinforcement Learning (GCRL) algorithms are built upon Hindsight Experience Replay (HER), which densifies rewards through hindsight replay and leverages historical goal-achieving information to construct a learning curriculum. However, when the task is characterized by a non-Markovian reward (NMR), whose computation depends on multiple steps of states and actions, HER can no longer densify rewards by treating a single encountered state as the hindsight goal. The lack of informative rewards hinders policy learning, resulting in rolling out failed trajectories. Consequently, the replay buffer is overwhelmed with failed trajectories, impeding the establishment of an applicable curriculum. To circumvent these limitations, we deviate from existing HER-based methods and propose an on-policy GCRL framework, GCPO, which is applicable to both multi-goal Markovian reward (MR) and NMR problems. GCPO consists of (1) Pre-training from Demonstrations, which pre-trains the policy to possess an initial goal-achieving capability, thereby diminishing the difficulty of subsequent online learning. (2) Online Self-Curriculum Learning, which first estimates the policy's goal-achieving capability based on historical evaluation information and then selects progressively challenging goals for learning based on its current capability. We evaluate GCPO on a challenging multi-goal long-horizon task: fixed-wing UAV velocity vector control. Experimental results demonstrate that GCPO is capable of effectively addressing both multi-goal MR and NMR problems.

## 1 Introduction

Multi-goal problems are ubiquitous in real-world applications, such as controlling robotic arms to grasp objects at any location on a table [13, 16], and operating fixed-wing Unmanned Aerial Vehicles (UAVs) to navigate towards any specified velocity vector [8, 34], etc. To address the challenge of automatically learning policies capable of achieving and generalizing across a range of diverse goals [47], Goal-Conditioned Reinforcement Learning (GCRL) [48, 37] has emerged as a prominent area of research. Serving as a generalization of standard Reinforcement Learning (RL) [61], GCRL learns goal-conditioned policies [55] through interactions within multi-goal environments [48].

In existing GCRL algorithms, Hindsight Experience Replay (HER) [3] plays a pivotal role in facilitating the learning of goal-conditioned policies. First, HER enhances sample efficiency by replacing the desired goals of failed trajectories with achieved states, thereby providing more informative rewards for policy learning [70, 11]. Second, HER contributes to creating a curriculum that enables the policy to progressively master challenging goals [40, 69, 47]. This process involves fitting the current policy's goal-achieving capability with historical goal-achieving information and selecting goals of appropriate difficulty for the current policy learning. Due to the reliance on replay buffers [72], current research on GCRL predominantly focuses on off-policy RL approaches [51, 25].

---

*Co-corresponding authors: kelele.xu@gmail.com, dingbo@nudt.edu.cn. [†] Equal contribution.

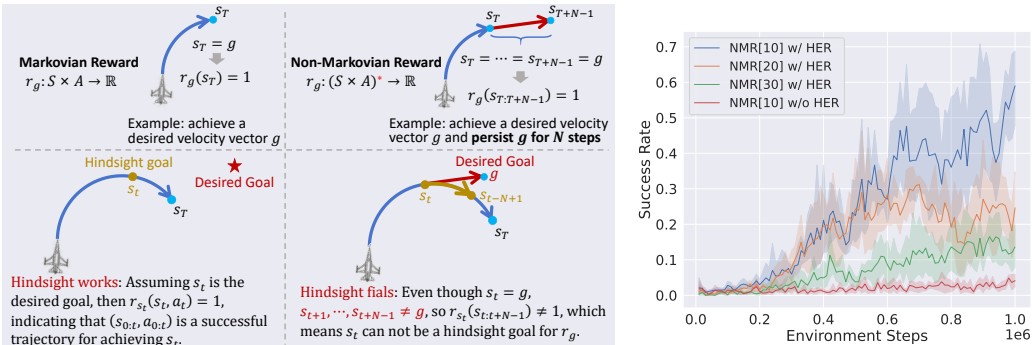

Figure 1: **Left:** Illustrations of HER on Markovian reward (MR) and non-Markovian reward (NMR) problems. **Right:** Performance of HER on MR and NMR problems. NMR[x] represents the NMR depends on the last x consecutive states and actions. Results come from experiments on an easy version (Appendix A.7) of fixed-wing UAV velocity vector control task over 5 random seeds.

Despite its widespread success, HER is subject to a limitation: the reward function must be able to determine goal achievement based solely on the current state, which implies that the reward function must be Markovian [2]. In real-world applications, many reward functions depend on multiple steps of states and actions, i.e., rewards are non-Markovian [24, 1]. For instance, determining whether an UAV has stably achieved a goal based on a sequence of states, or calculating a penalty for control oscillation based on a sequence of actions [33, 8]. For non-Markovian reward (NMR) problems, on one hand, it is not feasible to treat a single state from the sampled trajectory as a hindsight goal; on the other hand, it is challenging for the trajectory obtained through exploration to include the special transition sequence that satisfies the specific NMR. In Fig. 1, we present a specific case to elucidate the NMR problem and illustrate why HER fails to address NMR, and corresponding experimental results to support this explanation. It is evident that as the NMR relies on longer state-action sequences, the performance of HER gradually deteriorates, ultimately becoming indistinguishable from the performance when HER is not employed. This limitation prevents HER from densifying the reward. The lack of informative rewards hinders policy learning, which further leads to rolling out failed trajectories. Consequently, the replay buffer is overwhelmed with failed trajectories, obstructing the creation of a reasonable curriculum.

In light of the aforementioned limitations of HER in addressing NMR problems, we propose a novel on-policy [62, 57] GCRL framework, termed the Goal-Conditioned Policy Optimization (GCPO) framework, which deviates from existing HER-based off-policy approaches. GCPO comprises: (1) **Pre-training from Demonstrations**: This phase involves leveraging demonstrations to pre-train the policy offline, equipping it with an initial capability to achieve goals before online learning begins, thereby diminishing the difficulty of subsequent online learning. (2) **Online Self-Curriculum Learning**: This phase involves periodically evaluating the policy, estimating its goal-achieving capability, and optimizing it through online learning with progressively challenging goals sampled based on its current capability. It is important to note that GCPO is not specifically designed for NMR problems; instead, it is a general on-policy GCRL framework applicable to both multi-goal Markovian reward (MR) and NMR problems.

To evaluate GCPO, we conduct experiments on the Velocity Vector Control (VVC) task of fixed-wing UAVs. The VVC task represents a typical multi-goal problem that can be formulated as both MR and NMR, and it also requires long interaction sequences, categorizing it as a challenging long-horizon problem [28]. The complexity inherent in the VVC task provides a rigorous testbed for evaluating the efficacy of GCPO. Experimental results indicate that GCPO effectively addresses multi-goal problems, both MR and NMR. Our contributions are summarized as follows:

- We analyze the reason why existing GCRL algorithms, which rely on HER as a central component, fail in handling NMR problems and validate this finding through experiments.

- We propose an on-policy GCRL framework, GCPO, which incorporates pre-training from demonstrations and online self-curriculum learning to address both MR and NMR problems.

- We evaluate the performance of GCPO on the challenging VVC task, demonstrating its effectiveness in solving both MR and NMR problems. Additionally, we conduct ablation studies to analyze the influence of the two components and hyper-parameters on learning.

## 2   Related Work

**Goal-conditioned reinforcement learning**. Many prior works assume access to a goal-conditioned reward function [15, 73] and view GCRL as a reward-driven, multi-task learning problem [68]. Existing GCRL methods are predominantly based on HER, situating them within the off-policy RL domain [61]. Pitis et al. [47] summarize existing methods and propose a common off-policy GCRL framework, which alternates between collecting experience and optimizing policies. During the policy optimization phase, the hindsight replay method is utilized to relabel transitions sampled from the replay buffer, thereby increasing informative rewards in the training data. When collecting experience, some heuristics or learning methods are employed to select appropriate goals that assist in improving the policy.

Selecting appropriate goals involves (1) several heuristics for discovery, including reward relevance [4, 65], diversity [14, 10], coverage [38, 6], difficulty [32, 43], etc. (2) selecting goals of appropriate difficulty based on the capability of the current policy, which essentially craft objectives that try to optimize for learning progress. These methods fit the current policy's goal-achieving capability based on the trajectories stored in the replay buffer and then sampling goals of appropriate difficulty for online learning according to this capability. For instance, RIG [40] samples goals directly from the distribution of achieved goals, DISCERN [69] samples uniformly on the support set of the distribution of achieved goals, and MEGA [47] uses inverse probability weighting sampling [32] on the distribution of achieved goals to samples goals that the current policy can achieve but not well.

The difference between our method and the aforementioned methods lies in the learning framework: our method is on-policy, whereas the existing methods, all based on HER, are off-policy. This difference manifests in two key aspects: First, our method does not include a component like HER, which cannot be used for solving NMR problems. To achieve an effect similar to HER's enhancement of informative rewards during learning, we design a component that pre-trains the policy with demonstrations, making it suitable for on-policy RL. Second, our method employs off-policy evaluation [63, 64] to estimate the current policy's goal-achieving capability, rather than estimating this capability based on the trajectories in the replay buffer. Table 1 summarizes the similarities and differences between GCPO and existing GCRL methods.

Table 1: Comparison between HER-based methods and GCPO. $p_{ag}$ refers to the distribution of achieved goals, $supp(\cdot)$ refers to the support set of a distribution, $\mathcal{U}(\cdot)$ refers to the uniform distribution on a set, and $IPW(\cdot)$ refers to the inverse probability weighting [32].

| Method | | Type of RL | Applicable Reward Types | Method for increasing informative rewards | Method for sampling goal | |
|---|---|---|---|---|---|---|
| | | | | | estimate $p_{ag}$ | sample distribution |
| HER-based (NeurIPS2017) | +RIG (NeurIPS2018) +DISCERN (ICLR2019) +MEGA (ICML2020) | off-policy | MR | hindsight replay | replay buffer | $p_{ag}$ $\mathcal{U}[supp(p_{ag})]$ $IWP(p_{ag})$ |
| | GCPO | on-policy | MR, **NMR** | pre-train from demostrations | off-policy evaluation | $p_{ag}, \mathcal{U}[supp(p_{ag})],$ $IWP(p_{ag}), \cdots$ |

**Non-Markovian rewards.** Abel et al. [2] underscore the necessity of NMRs by demonstrating the existence of environment-task pairs for which no MR function can realize the task. Moreover, numerous exploration strategies implicitly depend on NMRs [35, 45], and studies considering non-Markovian discount factors [19, 58] can also be interpreted as special forms of NMRs [46]. The fundamental approach for addressing NMRs involves augmenting the state space to render the reward Markovian [24]. Various techniques have been proposed to achieve this, such as Reward Machines [9] and the Split-MDP [1]. Nevertheless, the expanded reward state space may grow exponentially with the number of acceptable policies and could incorporate an infinite number of simple reward functions [1]. Our work diverges from the aforementioned research by presenting a general GCRL framework, rather than a specialized approach for NMR problems. Unlike existing GCRL methods that depend on HER, which is not applicable to NMR problems, our framework can be applied to

Figure 2: The overall GCPO framework.

both NMR and MR problems. Additionally, our framework is compatible with techniques designed to address NMR challenges, as demonstrated in Appendix F.

## 3 Methodology

GCPO is designed as an on-policy GCRL framework. We draw upon the key insights from existing HER-based GCRL methods that have led to their success and incorporate two critical components into the GCPO framework: pre-training from demonstrations and online self-curriculum learning. The overall GCPO framework is depicted in Fig. 2 and a practical implementation of GCPO is detailed in Algorithm 1.

### 3.1 Preliminaries

GCRL can be described by goal-augmented MDP [37] $M = \langle \mathcal{S}, \mathcal{A}, \mathcal{T}, r, \gamma, \mathcal{G}, p_{dg}, \phi \rangle$, where $\mathcal{S}, \mathcal{A}, \gamma, \mathcal{G}$ and $p_{dg}$ denote the state space, action space, discount factor, goal space and desired goal distribution of the environment, respectively. $\mathcal{T} : \mathcal{S} \times \mathcal{A} \rightarrow \mathcal{P}(\mathcal{S})$ is the transition function, where $\mathcal{P}(\mathcal{X})$ denote the probability distribution over a set $\mathcal{X}$. $r$ is the goal-conditioned reward function. It can be both Markovian $r = \{r_g | r_g : \mathcal{S} \times \mathcal{A} \rightarrow \mathbb{R}, g \in G\}$ and non-Markovian $r = \{r_g | r_g : (\mathcal{S} \times \mathcal{A})^* \rightarrow \mathbb{R}, g \in G\}$. $\phi : \mathcal{S} \rightarrow \mathcal{G}$ is a tractable mapping function that maps the state to a specific goal. The objective of GCRL is to reach goals via a goal-conditioned policy $\pi : \mathcal{S} \times \mathcal{G} \rightarrow \mathcal{P}(\mathcal{A})$ that maximizes the expectation of the cumulative rewards over the desired goal distribution $J(\pi) = \mathbb{E}_{a_t \sim \pi(\cdot|s_t,g), g \sim p_{dg}, s_{t+1} \sim \mathcal{T}(\cdot|s_t,a_t)} \left[ \sum_t \gamma^t r_g(\cdot) \right]$. Additionally, previous works [47, 37] identify two common definitions of goals: Achieved goal, which refers to the goal accomplished by the policy in the current state. The notation $p_{ag}$ denotes the distribution of achieved goals. Behavioral goal, which represents the specific task that is targeted for sampling within a rollout episode [37].

### 3.2 Pre-Training from Demonstrations.

GCRL encounters more substantial exploration challenges compared to standard RL due to the inclusion of an additional goal space [30]. Pre-training from demonstrations is primarily designed to facilitate biased exploration [50]. Specifically, the policy can be pre-trained with Imitation Learning (IL) [74] or goal-conditioned IL [13, 25] on demonstrations. The pre-training provides the policy with a warm start [60, 71], which refers to an initial ability to achieve some of the desired goals. Pre-training is vital for on-policy RL as it enhances informative rewards during online learning. Without such informative rewards, the policy would struggle to acquire any meaningful knowledge or skills [61]. Through subsequent online learning, the policy can effectively discern when it is more advantageous to adhere to states and actions from the demonstration trajectories or to explore superior alternatives.

In our implementation of GCPO, we utilize **Behavioral Cloning** (BC) [49] for pre-training the policy, as indicated in line 2 of Algorithm 1. For the demonstration $\mathcal{D}_E$, the policy is learned by optimizing a supervised loss function to maximize the likelihood of expert actions [54]

$$\mathcal{L}(\theta) = -\mathbb{E}_{(s,a) \sim \mathcal{D}_E}[\log \pi_\theta(a|s)]. \tag{1}$$

---

**Algorithm 1** Goal-Conditioned Policy Optimization (GCPO) framework

---

**Require:** demonstrations $\mathcal{D}_E$, distribution of desired goal $p_{dg}$, goal weight discount factor $\kappa$, online evaluation budget $N$, probability transform function $f$

**Ensure:** $\pi_\theta(\cdot|s, g)$

1: Initialize goal-conditioned policy $\pi_\theta(\cdot|s, g)$, goal buffer $B_g$ which stores tuples of achieved goals in evaluation and their corresponding weight $(g, w_g)$
2: pre-train $\pi_\theta(\cdot|s, g)$ by Eq. 1             ▷ pre-train policy
3: **while** Not converge **do**
4:     **for all** $(g, w_g)$ in $B_g$ **do**
5:        $w_g \leftarrow \kappa \cdot w_g$            ▷ decay weight of historically achieved goals
6:     **end for**
7:     sample $N$ goals, $g_1, g_2, \ldots, g_N$ uniformly from $p_{dg}$       ▷ online policy evaluation
8:     **for all** $g$ in $g_1, g_2, \ldots, g_N$ **do**
9:        **if** $\pi_\theta$ finishes $g$ successfully **then**
10:          add $(g, 1.0)$ to $B_g$
11:        **end if**
12:     **end for**
13:     estimate $p_{ag}$ with GMM on $B_g$               ▷ estimate $p_{ag}$
14:     $\mathcal{D} \leftarrow \varnothing$                     ▷ roll-out samples
15:     **while** Not collect enough online samples **do**
16:        sample a goal $g$ from Eq. 2
17:        sample a trajectory $\tau$ by $\pi_\theta$ on $g$
18:        append $\tau$ to $\mathcal{D}$
19:     **end while**
20:     update $\pi_\theta$ by Eq. 3 on $\mathcal{D}$              ▷ update policy
21: **end while**

---

### 3.3 Online Self-Curriculum Learning

In GCRL, selecting goals that match the current policy's capabilities is crucial for effective learning [20, 12, 5, 11]. To address this, we design an online self-curriculum learning mechanism that autonomously constructs a curriculum, generating behavioral goals that are incrementally more difficult than those the policy is currently capable of achieving during training. Specifically, online self-curriculum learning consists of three processes: (1) Estimating the current policy's goal-achieving ability, $p_{ag}$. This can be done through methods such as online evaluation or off-policy evaluation (OPE) [63, 64]. (2) Setting or learning a probability transform function $f : \mathcal{P} \times \mathcal{P} \to \mathcal{P}$, followed by sampling progressively challenging behavioral goals based on $f(p_{ag}, p_{dg})$ for online learning. (3) Conducting online RL learning with behavioral goals sampled in the second part, facilitating the agent's progression towards more challenging goals. In our implementation of GCPO:

**Gaussian Mixture Model** (GMM) [53] is employed to estimate $p_{ag}$, as detailed in lines 4-13 of Algorithm 1. During the online self-curriculum learning, the policy is periodically evaluated. In each evaluation, the policy is evaluated with $N$ goals sampled from $p_{dg}$. Information about the achieved goals, along with an initial weight of 1.0, is stored in a goal buffer $B_g$. As the online self-curriculum learning proceeds, the weight of historically achieved goals is reduced by a factor $\kappa$. Ultimately, a GMM is used to estimate $p_{ag}$ based on the data in $B_g$ (The specific calculation can be referred to Appendix B.).

**Maximum Entropy Gain Exploration** (MEGA) [47] is utilized as the probability transform function $f$, as indicated in lines 14-19 of Algorithm 1. The core idea behind MEGA is to encourage exploration in sparsely explored areas of the achieved goal distribution. In discrete settings, inverse probability weighting [32] can be applied to sample goals from $p_{ag}$. A goal $g$ is chosen with the probability given by

$$\big[f_{MEGA}(p_{ag}, p_{dg})\big](g) = \frac{\frac{1}{p_{ag}(g)}}{\sum_{p'} \frac{1}{p_{ag}(g')}}. \tag{2}$$

In continuous settings, a generate and test strategy [42, 47] is employed for goal sampling. Specifically, $M$ goals $\{g_i\}_{i=1}^M$ are randomly sampled from $supp(p_{ag})$, the support set of $p_{ag}$, and the goal with the

minimum density under $f_{MEGA}(p_{ag}, p_{ag})$ is selected: $g = \arg\min_{g_i} \left( \left[ f_{MEGA}(p_{ag}, p_{ag}) \right](g_i) \right)$. This approach biases the sampling towards goals that are less likely under the current achieved goal distribution, promoting exploration in under-explored regions.

**KL-regularized RL** [67, 66, 7, 27] is employed as the on-policy RL algorithm to optimize the policy, as indicated in line 20 of Algorithm 1. To prevent catastrophic forgetting of latent skills and to continuously improve exploration during the RL fine-tuning phase [7], the policy $\pi_\theta$ is initially set to the pre-trained policy $\pi_{\theta_0}$ and is then fine-tuned by maximizing the following objective:

$$J_{kl}(\pi_\theta) = \mathbb{E}\big[\sum_t \gamma^t \big(r - \lambda log(\frac{\pi_\theta(a_t|s_t)}{\pi_{\theta_0}(a_t|s_t)})\big)\big], \tag{3}$$

where $r$ can represent both Markovian $r(s_t, a_t)$ and non-Markovian $r(s_{0:t}, a_{0:t})$ rewards, and $\lambda \in [0, 1]$ controls the strength of the KL regularization. Optimizing Eq. 3 is analogous to optimizing the original RL objective within the log-barrier of $\pi_{\theta_0}$, and can be viewed as a trust-region-style [56] learning objective [36].

## 4 Experiments

### 4.1 Experimental Setups

**RL environment.** Experiments are conducted on the **Fixed-Wing UAV Velocity Vector Control** (VVC) task [26], which is a representative multi-goal problem. The VVC task is characterized by a long horizon, with the average length of demonstrations exceeding 280 steps (detailed in Appendix A.5). Even for well-trained policies, the average number of steps required to achieve a goal is over 100, and more challenging goals can demand upwards of 300 steps to achieve [27]. This exceeds the horizon typically used in most previous studies [41, 47, 28, 59]. Additionally, the rewards for VVC can be designed as either MR or NMR, thereby accommodating a range of real-world requirements and complexities. The specifics of the VVC environment setup are detailed in Appendix A. Thus, VVC presents a challenging multi-goal, long-horizon problem that poses significant difficulties for policy learning. Standard SAC+HER [3] and PPO [57] are unable to solve the VVC task, as demonstrated in Appendix A.6, further highlighting the task's complexity. To our knowledge, previous research on NMR algorithms has primarily been tested in simpler environments such as multi-arm bandits [24] and grid worlds [52, 24, 2, 1, 31]. Our work is the first to evaluate the performance of algorithms on complex, real-world NMR problems. Additionally, to demonstrate the broad applicability of GCPO, we conduct experiments on the commonly used RL environments **Reach** and **PointMaze**. The corresponding results and analysis can be found in Appendix D.

**Demonstrations.** We collect a demonstration set $\mathcal{D}_E$, also denoted as $\mathcal{D}_E^0$, with a PID controller (detailed in Appendix A.5). Subsequently, we employ the IRPO algorithm [27], which iteratively optimizes policies and demonstrations, to generate $\mathcal{D}_E^1, \mathcal{D}_E^2, \mathcal{D}_E^3$. Table 2 presents the quantity and quality of these four demonstration sets. The '#traj' column represents the number of demonstrations contained within each demonstration set, while 'traj length' indicates the average length of demonstrations in the set. A shorter demonstration length suggests a faster completion of the corresponding goal, indicative of higher demonstration quality. It can be observed that the demonstration quantity and quality of $\mathcal{D}_E^0, \mathcal{D}_E^1, \mathcal{D}_E^2, \mathcal{D}_E^3$ increase sequentially.

### 4.2 Main Results

We compare GCPO with several baselines on the VVC task under different demonstration conditions, including (1) SAC [29] + HER + MEGA, which is a strong baseline in GCRL; (2) BC, a fundamental yet effective IL algorithm; (3) GCPO without pre-training, which corresponds to PPO + self-curriculum; (4) GCPO without self-curriculum, which corresponds to BC + KL-regularized RL. GCPO itself is equivalent to BC + KL-regularized RL + self-curriculum. Table 2 reports the performance of GCPO and the baselines on NMR, and Fig. 3 visualizes the learning progression on both NMR and MR, as well as the final learned policy of GCPO.

**GCPO is applicable to both MR and NMR problems.** Table 2 shows that GCPO outperforms all baselines on NMR, with SAC+HER+MEGA achieving only 20% of GCPO's performance. This demonstrates the limitations of HER in addressing NMR problems and highlights the superiority of GCPO in these contexts. Furthermore, Fig. 3a illustrates the learning progression of GCPO for

Table 2: Comparison between GCPO and baselines on NMR. The mean and variance of % success rates are presented over 5 random seeds. Optimal values are highlighted in bold, and sub-optimal values are underlined.

| Demonstration | | | SAC + HER + MEGA | BC | GCPO w/o pre-training | GCPO w/o self-curriculum | GCPO |
| --- | --- | --- | --- | --- | --- | --- | --- |
| notation | #traj | traj length | | | | | |
| $\mathcal{D}_E^0$ | 10264 | 281.83±149.48 | | 17.08±0.57 | | 31.28±8.97 | **45.87±3.09** |
| $\mathcal{D}_E^1$ | 27021 | 119.64±47.55 | 8.32±1.86 | 36.54±1.97 | 0.04±0.03 | 43.49±3.85 | **49.12±1.67** |
| $\mathcal{D}_E^2$ | 34952 | 115.76±45.65 | | 41.79±0.44 | | 51.28±2.07 | **57.45±2.49** |
| $\mathcal{D}_E^3$ | 39835 | 116.56±47.62 | | 42.77±1.35 | | 53.51±3.18 | **59.90±1.78** |

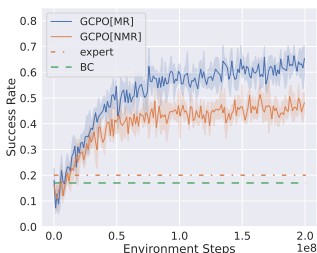

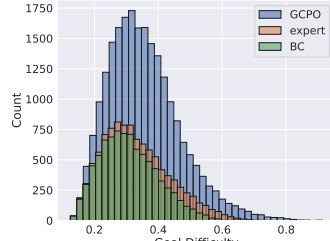

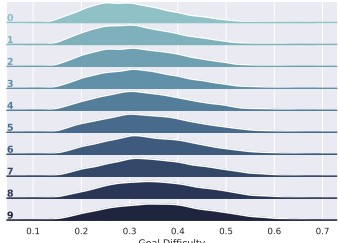

(a) Success rate on MR and NMR

(b) Histogram of achieved goals

(c) Distribution of goals from self-curriculum during learning

Figure 3: Main results of GCPO. 'expert' refers to the demonstrator that generates demonstrations. 'BC' refers to the pre-trained policy. Results are derived from experiments across 5 random seeds. For sub-figure (a), expert and BC are both evaluated in the NMR setting. For sub-figure (c), the vertical axis represents the training progress, where $0, 1, \cdots, 9$ correspond to $10\%, 20\%, \cdots, 100\%$ of the training progress, respectively.

both NMR and MR, showing that GCPO is effective in solving both types of problems. In summary, GCPO exhibits versatility and applicability across both MR and NMR problems.

**Pre-training is crucial for the success of GCPO.** As evidenced by the experiments in Table 2 for BC, GCPO w/o pre-training, and GCPO, it is observed that without pre-training, GCPO struggles to learn meaningful skills. However, a policy that is merely pre-trained, albeit with non-optimal performance, plays a crucial role in enabling GCPO to develop an effective policy. Fig. 3b provides a histogram of the achieved goals for the trained policies. It is evident that, even with a pre-trained policy that initially exhibits inferior performance compared to the demonstrator, GCPO's online self-curriculum learning facilitates significant improvement in the policy's performance, surpassing that of the demonstrator.

**Online self-curriculum facilitates the mastery of challenging goals.** Table 2 demonstrates that the application of self-curriculum within GCPO leads to an average 8.2% increase in policy performance compared to its absence. This enhancement is illustrated in Fig. 3c, which shows that the online self-curriculum mechanism systematically introduces more difficult goals into the learning progression as the policy gains proficiency. This mechanism effectively explains the advantages of self-curriculum for GCPO in mastering challenging goals.

## 4.3 Ablation Studies

In this section, we conduct ablation studies on the demonstration's quantity and goal distribution, analyze the sensitivity of GCPO to the parameters used for estimating $p_{ag}$, and compare the effectiveness of different self-curriculum methods.

### 4.3.1 Ablation on Quantity of Demonstrations

To illustrate the influence of the quantity of demonstrations on GCPO, we train GCPO with 10%, 50%, and 100% of $\mathcal{D}_E$ and present the performance of pre-trained policies and GCPO policies in Fig. 4a.

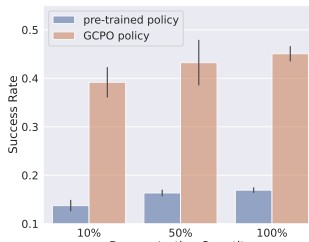 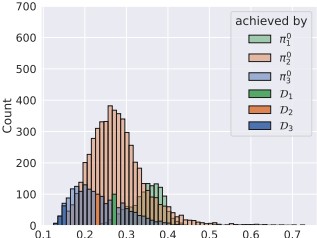 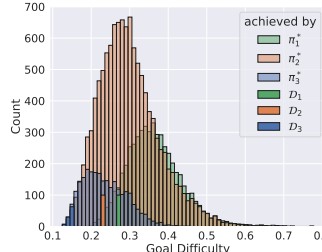

(a) Success rate of GCPO with different demonstration quantity

(b) Histogram of achieved goals of the pre-trained policy

(c) Histogram of achieved goals of GCPO policy

Figure 4: The influence of demonstration quantity and the distribution of goals covered by demonstrations on GCPO. $\mathcal{D}_1, \mathcal{D}_2, \mathcal{D}_3$ represent sets of demonstrations that are difficult, medium, and easy, respectively. The pre-trained policies obtained from $\mathcal{D}_1, \mathcal{D}_2$, and $\mathcal{D}_3$ are denoted as $\pi_1^0, \pi_2^0$, and $\pi_3^0$, respectively. The corresponding GCPO policies are denoted as $\pi_1^*, \pi_2^*$, and $\pi_3^*$, respectively. Results are derived from experiments across 5 random seeds.

Table 3: Performance of GCPO policies trained with different $N$ and $\kappa$. In our settings, $N = 32$ implies that the number of evaluations throughout the training is approximately equal to the number of goals obtained through discretizing the entire goal space during sampling demonstrations (detailed in Appendix A.5). The mean and variance of % success rates are shown over 5 random seeds. Optimal values are highlighted in bold, and sub-optimal values are underlined.

| (a) RIG | | | | (b) DISCERN | | | | (c) MEGA | | | |
|---|---|---|---|---|---|---|---|---|---|---|---|
| $\kappa$ \ $N$ | 32 | 96 | 320 | $\kappa$ \ $N$ | 32 | 96 | 320 | $\kappa$ \ $N$ | 32 | 96 | 320 |
| 0.9 | 46.28±1.10 | 47.22±1.51 | **49.03±1.54** | 0.9 | 46.92±3.54 | 47.20±2.49 | 49.36±1.91 | 0.9 | 42.58±1.69 | 45.87±3.09 | 48.62±2.35 |
| 0.99 | 47.23±0.88 | 46.49±2.97 | 47.62±1.34 | 0.99 | 45.43±1.52 | 48.59±3.94 | 48.18±2.51 | 0.99 | 43.35±1.00 | 45.06±1.30 | **49.21±2.23** |
| 0.995 | 46.14±4.79 | 47.75±1.70 | 48.13±2.01 | 0.995 | 47.06±2.51 | 47.36±1.98 | **50.08±1.07** | 0.995 | 43.56±0.63 | 46.42±2.42 | 46.19±4.27 |
| avg. | 46.55 | 47.15 | 48.26 | avg. | 46.47 | 47.72 | 49.21 | avg. | 43.16 | 45.78 | 48.01 |

As shown, a policy pre-trained with 10% of $\mathcal{D}_E$ achieves 81.12% of the performance of the policy pre-trained with 100% of $\mathcal{D}_E$, while a policy pre-trained with 50% of $\mathcal{D}_E$ achieves 96.53% of the performance. Similarly, a policy trained by GCPO with 10% of $\mathcal{D}_E$ achieves 86.95% of the performance of the policy trained by GCPO with 100% of $\mathcal{D}_E$, whereas a policy trained by GCPO with 50% of $\mathcal{D}_E$ achieves 95.94% of the performance. These results suggest that an increase in the quantity of demonstrations can enhance the performance of GCPO, yet the marginal gains diminish as the quantity of demonstrations grows. Furthermore, these results also indicate that GCPO can still perform well when only a relatively small number of demonstrations are available.

### 4.3.2 Ablation on Goal Distribution of Demonstrations

To demonstrate the influence of the distribution of goals covered by the demonstrations on GCPO, we collect three demonstration sets with significantly different goal difficulty distributions (detailed in Appendix A.4) and train GCPO with them. Fig. 4b presents the distribution of achieved goals of the pre-trained policies, while Fig. 4c depicts that of the GCPO policy.

It is evident that for both pre-trained policies and GCPO policies, their distributions of achieved goals are centered around the distribution of goals covered by the demonstrations. The reason for this is that the self-curriculum, starting with the distribution of achieved goals of the pre-trained policy, which is determined by the distribution of goals covered by the demonstrations, gradually expands the distribution of achieved goals. The preference for goals in the demonstrations thus influences the learning progression of GCPO, leading the policy learned by GCPO to also exhibit a similar preference for goals. This suggests that when preparing demonstrations for GCPO, it is preferable to sample goals and generate demonstrations as closely as possible to the desired goal distribution $p_{dg}$.

Furthermore, in Appendix E.1, we provide a more intuitive case and directly visualize the achieved goals in three-dimensional space, yielding the same conclusions as those from the above analysis.

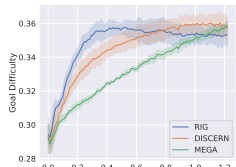 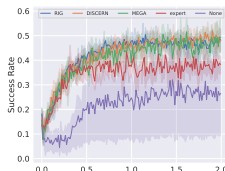 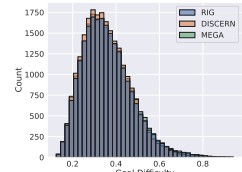 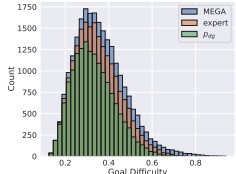

(a) Difficulty of goals sampled by self-curriculum methods

(b) Success rate of self-curriculum and non-curriculum methods

(c) Histogram of achieved goals of different self-curriculum methods

(d) Histogram of achieved goals of MEGA and non-curriculum methods

Figure 5: Analysis of the influence of different self-curriculum methods on the learning progression of GCPO, as well as a comparison between self-curriculum and non-curriculum methods. 'expert' and 'None' are two non-curriculum methods, where 'expert' refers to sampling goals from those that the demonstrator can achieve, and 'None' signifies directly sampling from $p_{dg}$. Results are derived from experiments across 5 random seeds.

### 4.3.3 Sensitivity to Parameters of Estimating $p_{ag}$

To evaluate the influence of the estimation of $p_{ag}$ on GCPO, we conduct experiments with GCPO across a range of values for $N$ and $\kappa$, employing various self-curriculum methods. The results are presented in Table 3.

The internal three-by-three layout of each sub-table reveals that policies with the highest performance tend to be situated at configurations where $N$ is larger and $\kappa$ is smaller. This trend suggests that under these conditions, the estimation of $p_{ag}$ is more precise. Ideally, as $N \to \inf$ and $\kappa \to 0$, the estimation of $p_{ag}$ could be perfectly fitted. when examining $N$ and $\kappa$ independently, the last row of each sub-table indicates that conducting more evaluations during online learning helps GCPO to obtain a well-performing policy, although this comes at the cost of additional computational resources. Conversely, the sum of each row of each sub-table shows no significant difference, implying that GCPO is less sensitive to the setting of $\kappa$.

In summary, from the perspective of estimating $p_{ag}$, to enhance the performance of GCPO, it is primarily advisable to increase the number of evaluations within the tolerable computational resource constraints.

### 4.3.4 Comparison on Different Self-Curriculum Methods

To demonstrate the influence of different self-curriculum methods on the learning progression and final policy of GCPO, we train GCPO with three distinct self-curriculum methods: RIG, DISCERN, and MEGA. The learning curve and the final policy performance are illustrated in Fig. 5.

Fig. 5a presents the curve of goal difficulty sampled during learning. It is noted that RIG rapidly samples more challenging goals, followed by DISCERN, and then MEGA. However, when considering the difficulty of the goals sampled at the final stage of learning, DISCERN and MEGA select harder goals than RIG. This observation suggests that RIG, DISCERN, and MEGA exhibit distinctly different learning progressions.

Fig. 5b depicts the trend of success rate during learning, and Fig. 5c and 5d present the histograms of achieved goals for the policies trained by self-curriculum and non-curriculum methods, respectively. By combining Figs. 5b and 5c, it is evident that there is no significant difference in performance between different self-curriculum methods, whether in the learning progression or in the final policy. In contrast, when combining Figs. 5b and 5d, it is clear that self-curriculum methods outperform non-curriculum methods in both the learning progression and the final policy performance.

In summary, within the GCPO framework, while different self-curriculum methods exhibit distinct learning progressions, there is no discernible difference in the final policy obtained. Moreover, the self-curriculum methods consistently outperform non-curriculum methods, highlighting the effectiveness of the self-curriculum mechanism in promoting goal-conditioned policy learning.

## 5 Conclusion and Limitations

In this paper, we propose an on-policy goal-conditioned reinforcement learning framework, GCPO, designed to address the limitations of existing methods in solving non-Markovian reward (NMR) problems. Through experimental evaluation on the fixed-wing Velocity Vector Control task, we demonstrate the effectiveness of GCPO in handling both Markovian reward (MR) and NMR problems.

Some limitations should be addressed in future work. Firstly, in the implementation of the two components within GCPO, we employ relatively simple methods, such as behavioral cloning and Gaussian mixture model. Whether the use of alternative methods could lead to more efficient learning and better-performing policies is yet to be further validated. Secondly, under the sparse reward setting, the successful training of GCPO relies on the pre-trained policy possessing a certain level of goal-achieving capability. Otherwise, if the policy achieves nothing, it becomes ineffective in establishing a self-curriculum. Lastly, although the GCPO framework does not have a component like HER that is unsuitable for solving NMR problems and thus capable of solving both MR and NMR problems, the specific implementation of GCPO as introduced in Section 1 does not explicitly incorporate components that are specifically designed to handle NMR problems. In Appendix F, we introduce a simple component to address NMR problems within GCPO and observe some effects. However, it is not clear whether integrating the most advanced methods for handling NMR problems within GCPO would lead to a more effective resolution.

## Acknowledgements

This work was supported by the National Key R&D Program of China (No. 2021ZD0112904).

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

# Appendix

## A  The Fixed-Wing Velocity Vector Control Task

The fixed-wing UAV's VVC task is to target its velocity vector to a target velocity vector.

### A.1  State, Action, and Goal Space

The state consists of pitch angle $\theta$, roll angle $\phi$, yaw angle $\psi$, flight path azimuth angle $\chi$, flight path elevator angle $\mu$, altitude $h$, roll angular velocity $p$, true airspeed $v$, and goal $(v_g, \mu_g, \chi_g)$. The action consists of $ail, ele, rud, pla$, which denotes the actuator position of the aileron, elevator, rudder, and power level actuator. The goal space is defined as $\mathcal{G} := [v_{min}, v_{max}] \times [\mu_{min}, \mu_{max}] \times [\chi_{min}, \chi_{max}]$. When the environment is reset, a goal $g := (v, \mu, \chi)$ is sampled randomly from $\mathcal{G}$.

### A.2  Transition

The action $(ail, ele, rud, pla)$ is sent to the Flight Dynamics Model (FDM) to get the next state with the F-16 model. The episode terminates when triggers one of the following two conditions: (1) if $v, \mu, \chi$ is close to $(v_g, \mu_g, \chi_g)$ within the tolerant error $\delta$ which is described in the Appendix A.3. (2) if does not trigger the first condition for $T_{max}$ steps.

### A.3  Reward Function

The Markovian reward function is defined as

$$r_g(s_t) = \begin{cases} 0, \; if \; d(\phi(s_t), g) < \delta \\ -1, \; else, \end{cases} \tag{4}$$

where $d(\phi(s_t), g) = w_v \frac{\|\vec{v}_t - \vec{v}_g\|_v}{\sigma_v} + w_d \frac{\|\vec{v}_t - \vec{v}_g\|_d}{\sigma_d}$, $w_v \in [0, 1], w_d \in [0, 1], w_v + w_d = 1.0$ are weight factors for velocity and direction, $\sigma_v, \sigma_d$ are scaling factors for velocity and direction, $\|.\|_v$ calculates the difference in modulus of two velocity vectors, $\|.\|_d$ calculates the difference in direction of two velocity vectors, and $\delta$ is a pre-defined tolerant error.

The non-Markovian reward function is defined as

$$r_g(s_{t-N_r+1}, \cdots, s_t) = \begin{cases} 0, \; if \; all \; d(\phi(s_{t'}), g) < \delta, \; t' \in [t - N_r + 1, t - N_r + 2, \cdots, t] \\ -1, \; else, \end{cases} \tag{5}$$

where $N_r$ is the horizon length that the NMR depends on.

### A.4  Goal Difficulty

In order to evaluate the quality of demonstrations in the following sections, we introduce the goal difficulty

$$d_v(g, v_0) = \alpha_v + (1 - \alpha_v) \frac{|v_g - v_0|}{|v_{max} - v_{min}|}, \tag{6}$$

where $\alpha_v \in [0, 1)$ is a base value of difficulty for $v$. And the same is for $d_\mu(g, \mu_0)$ and $d_\chi(g, \chi_0)$. Consequently, the difficulty of the goal is defined as

$$d(g, v_0, \mu_0, \chi_0) = d_v(g, v_0) \cdot d_\mu(g, \mu_0) \cdot d_\chi(g, \chi_0), \tag{7}$$

which describes the magnitude of changes in the UAV's state variables.

Based on Eq. 7, we sort all goals based on their difficulty and define the following three goal sets: the easy goal set, comprising the 100 simplest goals; the medium goal set, consisting of goals with difficulty values ranked between 3000 and 3100; and the difficult goal set, comprising goals with difficulty values ranked between 7000 and 7100.

### A.5  Demonstrations

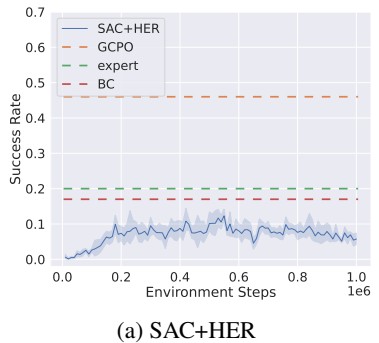
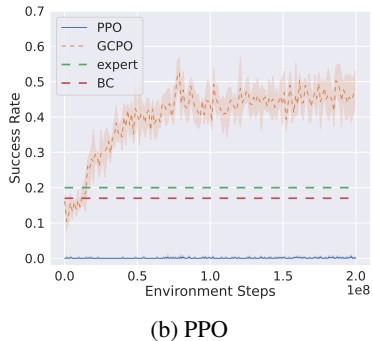

|   (a) SAC+HER   |   (b) PPO   |

Figure 6: Performance of SAC+HER and PPO on the VVC task. Results come from experiments on attitude control over 5 random seeds.

Table 5: Parameters used in Environment

| (a) Easy Version | | (b) Hard Version | |
| --- | --- | --- | --- |
| Parameter | Value | Parameter | Value |
| $v_{min}, v_{max}$ | **150, 250** | $v_{min}, v_{max}$ | **100, 300** |
| $\mu_{min}, \mu_{max}$ | **-10, 10** | $\mu_{min}, \mu_{max}$ | **-85, 85** |
| $\chi{min}, \chi{max}$ | **-30, 30** | $\chi{min}, \chi{max}$ | **-170, 170** |
| $T_{max}$ | 400 | $T_{max}$ | 400 |
| $w_v, w_d$ | 0.5, 0.5 | $w_v, w_d$ | 0.5, 0.5 |
| $\sigma_v, \sigma_d$ | 100, 180 | $\sigma_v, \sigma_d$ | 100, 180 |
| $\delta$ | 0.058 | $\delta$ | 0.058 |
| $N_r$ | 10 | $N_r$ | 10 |

A PID controller is used to sample trajectories. For convenience, the goal space is discretized with parameters listed in Table 4. Of the 50715 discretized goals, 10184 trajectories are successfully sampled with an average length of 282.01. These 10184 successful trajectories form the $\mathcal{D}_E$. $\mathcal{D}_E$ are imperfect: firstly, the quantity of demonstrations is limited with only about 20% goals successfully sampled; secondly, the quality of demonstrations is low as a well-trained policy can usually finish the goal within 150 steps.

Table 4: Attitude control

|   | min | max | $\Delta$ |
| --- | --- | --- | --- |
| $v$ | 100 | 300 | 10 |
| $\mu$ | -85 | 85 | 5 |
| $\chi$ | -170 | 170 | 5 |
| #goals: 50715 | | | |

## A.6 The multi-goal long-horizon problem

The VVC task represents typical multi-goal long-horizon problems. On the one hand, the average length of individual goals (without further division into sub-goals) in expert demonstrations exceeds 280 steps. Even for policies that have been trained thoroughly, the average steps to achieve a goal is over 100, with more challenging goals requiring more than 300 steps to complete, which is longer than the horizon used in most previous studies [41, 47, 28, 59]. On the other hand, when trained with classical RL algorithms, as illustrated in Fig. 6, neither the off-policy SAC+HER nor the on-policy PPO algorithms are able to solve the VVC task. This also indicates the challenges that the long horizon of VVC poses for GCRL algorithms.

## A.7 Environment Hyper-Parameters Details

In the specific experiments, we only used the easy version, listed by Table 5a, of the VVC task in Fig. 1 of Sec. 1. In all experiments within the Sec. 4, we utilized the hard version, listed by Table 5b, of the VVC task.

Table 6: Parameters used in BC

| (a) BC | | | (b) PPO | |
| --- | --- | --- | --- | --- |
| Parameter | Value | | Parameter | Value |
| l2_weight | 0 | | ent_coef | $10^{-2}$ |
| ent_weight | $10^{-2}$ | | gamma | 0.995 |
| batch_size | 4096 | | gae_lambda | 0.95 |
| epochs | 300 | | lr | $10^{-4}$ |
| | | | batch_size | 4096 |
| | | | train_steps | $2 * 10^8$ |
| | | | rollout_process_num | 64 |
| | | | n_steps | 2048 |
| | | | n_epochs | 5 |
| | | | use_sde | True |
| | | | normalize_advantage | True |

# B    Calculation of GMM

We employ GMM with weighted samples [21] to estimate $p_{ag}$. For a set of $N$ weighted samples $\{(x_i, w_i)\}$, where $x_i$ is the $i^{th}$ sample and $w_i$ is the weight of $x_i$, we want to find a Guassian Model density function with $M$ Guassian components $f(x) = \sum_{i=1}^{M} \pi_i N(x|\mu_i, \Sigma_i)$, where $N(x|\mu_i, \Sigma_i) = \frac{1}{(2\pi)^{D/2}|\Sigma_i|^{1/2}} \exp\{-\frac{1}{2}(x-\mu_i)^T \Sigma_i^{-1}(x-\mu_i)\}$ is the $i^{th}$ Guassian component, $D$ is the dimension of sample, $\mu_i, \Sigma_i, \pi_i$ are the mean vector, covariance matrix, and the weight of $N(\cdot|\mu_i, \Sigma_i)$, correspondingly.

Then, Expectation-Maximization algorithm is employed to estimate the corresponding parameters. In the Expectation Step, the new estimate of each sample corresponds to each Guassian components is calculated, $r_{ik} = \frac{\pi_k N(x_i|\mu_k, \Sigma_k)}{\sum_{j=1}^{M} \pi_j N(x_i|\mu_j, \Sigma_j)}$. In the Maximization Step, calculate the weight of Guassian component, $\pi_k = \frac{\sum_{i=1}^{N} w_i r_{ik}}{\sum_{j=1}^{M} \sum_{i=1}^{N} w_i r_{ij}}$, the mean vector, $\mu_k = \frac{\sum_{i=1}^{N} w_i r_{ik} x_i}{\sum_{i=1}^{N} w_i r_{ik}}$, and the covariance matrix, $\Sigma_k = \frac{\sum_{i=1}^{N} w_i r_{ik}(x_i - \mu_k)(x_i - \mu_k)^T}{\sum_{i=1}^{N} w_i r_{ik}}$.

# C    Implementation Details

The Imitation framework is utilized to implement BC algorithm with parameters listed in Table 6a, and the Stable Baselines3 framework for PPO with parameters listed in Table 6b. 128*128 fully connected network and the Tanh activation function are used for VVC.

As BC only learns a policy network, we add a warm-up for the value network at the beginning of online learning. When online learning begins, we first freeze the parameter of the policy and train the value network with online samples until it converges, then proceed with the normal RL training. For the parameter $\lambda$, $10^{-3}$ is utilized for all the experiments.

# D    Experiments on Reach and PointMaze

We conduct two sets of experiments to demonstrate the general applicability of GCPO.

## D.1    Experimental Setups

**Environments:** For the first set of experiments, we conduct evaluations on a customized PointMaze environment (PointMaze_Large_DIVERSE_G-v3) from Gymnasium-Robotics [17] within the Mujoco physics engine. The only modification we made to the environment is to expand the number of desired goals from 7 to 45, making our customized version of PointMaze more challenging than the original version. For the second set of experiments, we employ a customized Reach (PandaReach-v3) [23] task on the Franka Emika Panda robot physics engine. The only modification we made to the environment is to change the distance_threshold used to determine goal reaching from 0.05 to 0.01.

Table 7: Comparison between GCPO and baselines on Reach and PointMaze. The mean and variance of % success rates are presented over 5 random seeds. Optimal values are highlighted in bold, and sub-optimal values are underlined.

| Task | Reward | SAC + HER + MEGA | BC | GCPO |
|------|--------|------------------|-----|------|
| Reach | MR | **100.0±0.0** | 70.63±2.99 | **100.0±0.0** |
| | NMR | 0.72±1.34 | 10.52±11.70 | **80.26±17.01** |
| PointMaze | MR | **100.0±0.0** | 75.96±5.34 | 93.33±3.06 |
| | NMR | 4.17±0.93 | 22.8±3.71 | **47.50±8.06** |

Consequently, our customized version of the Reach task has a stricter criterion for determining goal arrival, making it more difficult than the original version of Reach.

**Reward Settings:** The original rewards for both the PointMaze and Reach tasks are Markovian. To evaluate the performance of our algorithm under different NMR settings, we design two distinct types of NMRs. For the PointMaze, the NMR we designed is: the task is considered successful only if, after the point reaches the goal, it moves away by at least a certain distance and then returns to the goal. For the Reach task, the NMR we designed is: the Panda robot must first pass through a specific waypoint before reaching the goal to be considered successful, and each goal has a different waypoint. Both of these settings strictly adhere to the definition of NMR, where the reward is defined by the states and actions over multiple steps.

**Demonstrations:** The demonstrations for PointMaze are sourced from Minari [18] (pointmaze-large-v1), while the demonstrations for Reach are generated by us, with reference to the PID controller as described in the official documentation [44].

## D.2 Results

We evaluate SAC+HER+MEGA, BC, and GCPO on the PointMaze and Reach tasks under both MR and NMR settings. Table 7 presents the success rates of these algorithms. It can be observed that under the MR settings, GCPO exhibits similar performance to SAC+HER+MEGA. However, under the NMR settings, where HER cannot be effective, the performance of GCPO is significantly better than that of SAC. Taking into account the performance of GCPO on the VVC task as illustrated in Section 4.2, we showcase the general applicability of GCPO across a variety of tasks.

## E The Impact of Demonstrations on GCPO Training

In this section, we conduct a detailed experimental analysis of the impact of the distribution of goals that Demonstrations can cover, demonstration quantity, and demonstration quality on the training of GCPO.

### E.1 Goal Distribution Covered by Demonstrations

To more intuitively illustrate the impact of the goal distribution in demonstrations on GCPO training, we select three subsets of demonstrations from $\mathcal{D}_E$ that are spatially distant from each other, comprising all $\chi = 10$ demonstrations for $\mathcal{D}_{[\chi=10]}$, all $\chi = 90$ demonstrations for $\mathcal{D}_{[\chi=90]}$, and all $\chi = 170$ demonstrations for $\mathcal{D}_{[\chi=170]}$.

It is evident that, for both the pre-trained policy and the GCPO policy, the achieved goals are distributed around the goal distribution of the demonstrations. This suggests that the goal distribution in the demonstrations biases the learning of both pre-training and GCPO, leading to policies that share a similar goal distribution as the demonstrations. Therefore, the best practice when preparing demonstrations is to make the distribution of goals covered by the demonstrations as closely resemble the desired goal distribution as possible.

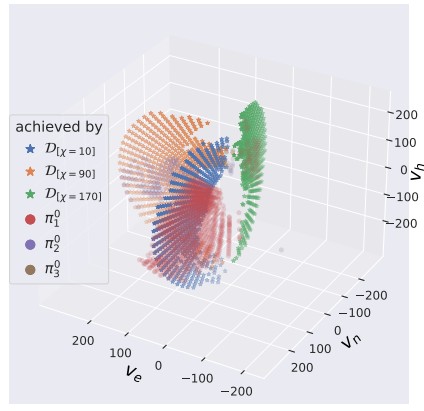 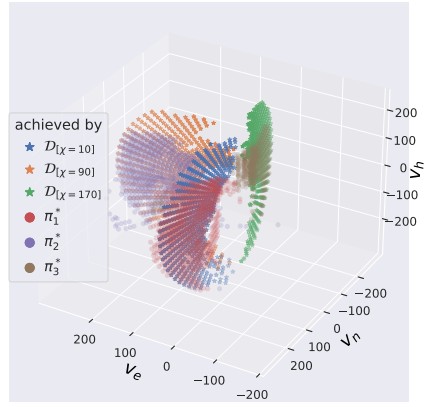

(a) Goals covered by demonstrations and goals achieved by pre-trained policy

(b) Goals covered by demonstrations and goals achieved by GCPO policy

Figure 7: Visualization of goals covered by demonstrations and achieved by policies. The pre-trained policies with $\mathcal{D}_{[\chi=10]}$, $\mathcal{D}_{[\chi=90]}$, and $\mathcal{D}_{[\chi=170]}$ are denoted as $\pi_1^0$, $\pi_2^0$, and $\pi_3^0$, respectively. The corresponding GCPO policies are denoted as $\pi_1^*$, $\pi_2^*$, and $\pi_3^*$, respectively.

Table 8: Performance of GCPO on Reach, PointMaze, and VVC with 1000 demonstrations, and more results on VVC with various demonstration quantities. The mean and variance of % success rates are presented over 5 random seeds.

| Task | Demo Quantity | Transition Quantity | BC | GCPO |
|---|---|---|---|---|
| Reach | 1000 | 82589 | 70.63±2.99 | 100.0±0.0 |
| PointMaze | 1000 | 1000000 | 75.96±5.34 | 93.33±3.06 |
| VVC + $10\%\mathcal{D}_E$ | 1000 | 294702 | 13.74±0.99 | 39.18±2.64 |
| VVC + $\mathcal{D}_{\chi=90}$ | 144 | 42177 | 1.18±0.38 | 3.34±0.61 |
| VVC + $\mathcal{D}_E$ | 10264 | 3106516 | 17.08±0.57 | 45.06±1.3 |

## E.2 Demonstration Quantity

We note that commonly used RL demonstration datasets typically consist of 1000 demonstrations [22]. Consequently, we evaluate the performance of GCPO on Reach, PointMaze, and VVC with 1000 demonstrations. In addition, we conduct an ablation study on demonstration quantity on VVC. The results are presented in Table 8.

Demonstrations: The source of demonstrations for PointMaze and Reach is elaborated in Section D. For GCPO, $10\%\mathcal{D}_E$ is employed, as detailed in Section 4.3.1. Additionally, for comparison purposes, we also present the performance of GCPO on the VVC task using $\mathcal{D}_{[\chi=90]}$, which comprises only 144 demonstrations with a flight path azimuth angle of 90, as described in detail in Appendix E.1, and using $\mathcal{D}_E$. The results are presented in Table 8. It can be observed that on the relatively simple PointMaze and Reach tasks, GCPO achieved nearly 100% success rate when using 1000 demonstrations. On the more challenging VVC task, the success rate with 1000 demonstrations reached 81.12% of the success rate achieved with 10264 demonstrations, while also being significantly higher than the success rate achieved with 144 demonstrations. These results indicate that across tasks of varying complexity, GCPO can achieve good performance with the use of 1000 demonstrations.

## E.3 Demonstration Quality

To obtain a demonstration set for comparison with $\mathcal{D}_E$, we generate trajectories for these goals using all the policies trained in our experiments. For a specific goal, we retain only the shortest trajectory. These generated trajectories are denoted as $\mathcal{D}'$.

Table 9: Demonstration quality of $\mathcal{D}_E$ and $\mathcal{D}'$.

| Demonstrations | Quantity | $I_l$ | $I_s$ |
|:---:|:---:|:---:|:---:|
| $\mathcal{D}_E$ | 10264 | 282.02±149.98 | 2.11±2.21 |
| $\mathcal{D}'$ | 10264 | 101.42±32.41 | 10.19±8.74 |

Table 10: Performance of GCPO on VVC with demonstrations of different demonstration qualities. $s(\pi)$ indicates the success rate of policy $\pi$. The mean and variance of % success rates are presented over 5 random seeds.

| Demonstrations | $s(\pi_{BC})\uparrow$ | $I_l(\pi_{BC})\downarrow$ | $I_s(\pi_{BC})\downarrow$ | $s(\pi_{GCPO})\uparrow$ | $I_l(\pi_{GCPO})\downarrow$ | $I_s(\pi_{GCPO})\downarrow$ |
|:---:|:---:|:---:|:---:|:---:|:---:|:---:|
| $\mathcal{D}_E$ | 17.08±0.57 | 241.72±81.36 | 1.97±1.97 | 45.87±3.09 | 133.86±53.24 | 6.84±5.60 |
| $\mathcal{D}'$ | 19.10±0.22 | 122.81±54.36 | 8.85±9.60 | 39.26±2.02 | 150.59±63.89 | 18.11±12.69 |

We use two metrics to measure demonstration quality: Trajectory length $I_l$. Since we employ $(-1, 0)$ sparse rewards, this implies that shorter trajectories yield a higher cumulative reward. Control smoothness $I_s$. In control problems, minimal control gains is expected to reduce wear on actuators. Hence, we refer to [39] to define the control smoothness. Trajectory length and control smoothness each describe certain characteristics of demonstrations from the distinct perspectives of reinforcement learning optimization and optimal control, respectively. Table 9 shows the demonstration quality of these two demonstration sets. It can be observed that: From an RL perspective, $\mathcal{D}'$ is of higher quality because the trajectories are shorter, leading to a higher expected cumulative reward. From a control perspective, $\mathcal{D}_E$ is better because the trajectories are smoother.

The performance of the BC policy $\pi_{BC}$ and the GCPO policy $\pi_{GCPO}$ trained on these two sets of demonstrations is shown in Table 10. The results reveal that $\pi_{BC}$ closely aligns with the demonstrations on both quality metrics, indicating that demonstration quality has a direct impact on BC. Additionally, the BC policy trained on $\mathcal{D}'$ has a slightly higher success rate, which we speculate is due to $\mathcal{D}'$ being more suitable for RL (the network architecture and training hyperparameters used to generate $\mathcal{D}'$ are the same as those for the BC policy). However, after the self-curriculum learning, the GCPO policy corresponding to $\mathcal{D}_E$ performs better and exhibits a shorter trajectory length. This suggests that the influence of demonstration quality on GCPO's online learning may not be as direct as pre-training, and further research is required to understand this relationship.

In summary, on one hand, it is challenging to define demonstration quality suitable for RL through a few metrics. On the other hand, demonstration quality does affect GCPO pre-training. How demonstration quality potentially influences the self-curriculum learning of GCPO remains an intriguing question for further exploration.

Furthermore, despite the aforementioned challenges, GCPO is capable of training well-performed policies from non-expert demonstrations. The intrinsic reason is that GCPO employs online learning to fine-tune pre-trained policies. Consequently, even if the demonstrations are non-expert and the pre-trained policies perform poorly, GCPO can still continuously optimize these policies through online learning.

In Section 4.2, although the average trajectory length of $\mathcal{D}_E^0$ reached 281.83, covering only 20.24% of goal space, the GCPO policy trained on it achieves a success rate of 45.87%, with an average trajectory length of 134.47. This comparison indicates that $\mathcal{D}_E^0$ consists of non-expert demonstrations. On the other hand, in contrast to $\mathcal{D}_E^3$, which covers 78.55% of the goal space with an average trajectory length of 116.56, $\mathcal{D}_E^0$ is only a quarter in size and has trajectories that are 2.42 times longer, implying a substantial decrease in its quantity and quality. Nonetheless, the GCPO policy trained on $\mathcal{D}_E^0$ achieves 76.58% of the success rate of the policy trained on $\mathcal{D}_E^3$.

# F   Extending GCPO with A Method for Solving NMR Problems

To demonstrate the effects of integrating a method for handling NMR problems into GCPO, we extend GCPO with a basic method for addressing NMR problems, which involves extending the input

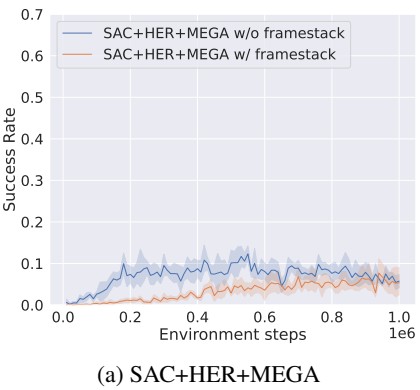
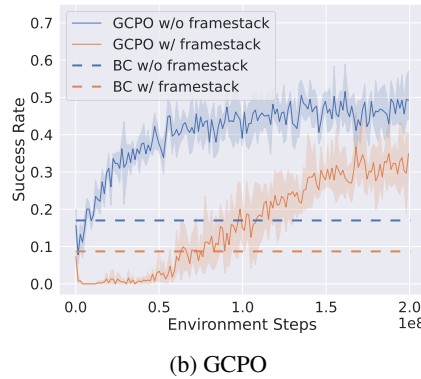

(a) SAC+HER+MEGA                                    (b) GCPO

Figure 8: Success rate of SAC+HER+MEGA and GCPO extended with consecutive states. 'w/ framestack' indicates that the policy input is extended with consecutive states, whereas 'w/o framestack' denotes that the input remains the current state.

of the policy with consecutive states to make the reward Markovian [24]. We evaluate the extended GCPO on the VVC task and show the result in Fig. 8.

It is evident that, regardless of whether it is the pre-trained policy or the GCPO policy or the SAC+HER+MEGA policy, expanding the input of the policy leads to a certain degree of performance degradation. We conjecture that this is likely due to the expansion of the policy's search space, which makes the original number of demonstrations and online learning resources relatively insufficient, indirectly increasing the difficulty of solving the policy. Therefore, although expanding the input is theoretically beneficial for addressing NMR problems, in practice, it also requires a balance between the increased difficulty of the problem and the available resources.

## G  Societal Impacts

This paper introduces a general on-policy goal-conditioned reinforcement learning framework, GCPO, which is capable of addressing both Markovian and non-Markovian reward problems. This flexibility makes GCPO particularly well-suited for a wide range of practical applications that rely on non-Markovian rewards, such as robotic arm control with stringent stability requirements and drone control that demands steady goal achievement. However, in practical settings, it is imperative to pay close attention to the evaluation of the GCPO policy, carefully analyzing the disparities between the policy's achieved goal distribution and the desired goal distribution of the actual task. Allowing the policy to attempt goals beyond its capabilities in a production environment could result in hardware failures, damage, and other problems.

