# OpenReview forum: "Goal-Conditioned On-Policy Reinforcement Learning"
_NeurIPS.cc/2024/Conference — NeurIPS 2024 poster_

### Official Review · Reviewer_aw7R · 2024-07-01

**Soundness:** 2
**Presentation:** 3
**Contribution:** 2
**Rating:** 7
**Confidence:** 3

**Summary:**

After rebuttal

The authors have mostly addressed my minor concerns. I recommend acceptance.

------


This paper aims to improve goal-conditioned RL. Two problems with prior methods (e.g. HER) are discussed, namely (a) their inability to cope with non-markovian rewards; and (b) the necessity of using an off-policy algorithm. This work then goes on to introduce a method that is applicable to on-policy algorithms *and* NMR problems. This method first uses demonstrations to pre-train a policy to achieve the goals sometimes. Then, it uses a curriculum-based approach to select harder and harder goals (but still within the agent's capability). They demonstrate on a UAV problem with NMR, they outperform other methods, notably SAC + HER.

**Strengths:**

1. Having a good on-policy GCRL algorithm is a great thing to shoot for, and the problem is useful.
2. Solving the problem of existing methods not being applicable to NMR domains is beneficial.
3. The set of baseline algorithms/ablations makes sense, and most of the obvious things to compare against have been included.

**Weaknesses:**

1. It would be nice to have experiments on another domain, to illustrate the general applicability of your algorithm.
	1. In particular, having results of the baseline algorithms (and GCPO) in an MR domain would be very helpful.
2. Relatedly, having results for e.g. SAC + HER when doing the same thing as in appendix A.4 would be helpful.

**Questions:**

- Could you run GCPO with only a 1000 demonstrations?

**Limitations:**

1. Requiring demonstrations is a relatively strong limitations, as e.g. HER does not. Is there any way around this requirement?

---

> ### Author Rebuttal · Authors · 2024-08-07
>
> We address the concern about GCPO performance in more RL domains (W1) in Global Author Response, and the other concerns below.
>
> > Q1: Could you run GCPO with only a 1000 demonstrations?
>
> We evaluate the performance of GCPO with 1000 demonstrations on the Reach, PointMaze, and VVC tasks.
>
> **Demonstrations**: The source of demonstrations for PointMaze and Reach is elaborated in the Global Author Response. For GCPO, 10% $\mathcal{D}_E$ is employed, as detailed in Section 4.3.1.
>
> Additionally, for comparison purposes, we also present the performance of GCPO on the VVC task using $\mathcal{D}_{[\chi=90]}$, which comprises only 144 demonstrations with a flight path azimuth angle of 90, as described in detail in Appendix A.3, and using $\mathcal{D}_E$. The results are presented in the following table:
>
> |Task|Demo Quantity|Transition Quantity|BC|GCPO|
> |:---:|:---:|:---:|:---:|:---:|
> |Reach|1000|82589|70.63±2.99|100.0±0.0|
> |PointMaze|1000|1000000|75.96±5.34|93.33±3.06|
> |VVC + 10% $\mathcal{D}_E$|1000|294702|13.74±0.99|39.18±2.64|
> |VVC + $\mathcal{D}_{[\chi=90]}$|144|42177|1.18±0.38|3.34±0.61|
> |VVC + $\mathcal{D}_E$|10264|3106516|17.08±0.57|45.06±1.3|
>
> **Performance**: It can be observed that on the relatively simple PointMaze and Reach tasks, GCPO achieved nearly 100% success rate when using 1000 demonstrations. On the more challenging VVC task, the success rate with 1000 demonstrations reached 81.12% of the success rate achieved with 10264 demonstrations, while also being significantly higher than the success rate achieved with 144 demonstrations. These results indicate that **across tasks of varying complexity, GCPO can achieve good performance with the use of 1000 demonstrations.**
>
> > L1: Requiring demonstrations is a relatively strong limitations, as e.g. HER does not. Is there any way around this requirement?
>
> We answer this question from two perspectives: Firstly, while GCPO's training relies on demonstrations, the quantity and quality of these demonstrations do not need to be high. GCPO is capable of learning well-performed policies from non-expert demonstrations. Secondly, for complex tasks, methods like SAC and HER are also unable to learn from scratch; leveraging demonstrations to assist learning is a more mainstream approach. We will elaborate on these points in the following discussion.
>
> On one hand, although GCPO relies on demonstrations, it is capable of learning well-performed policies from non-expert demonstrations. Please refer to the response to Q2 in the Global Author Response for a detailed analysis.
>
> On the other hand, for complex tasks, methods such as SAC, HER, etc., also struggle to learn well-performed policies from scratch. For instance, on the VVC task, the performance of policies obtained using SAC+HER+MEGA is shown in the table below, where it can be observed that this method barely learn any capability to complete tasks. In other complex tasks, such as StarCraft [1], Minecraft [2], ObjectGoal Navigation [3], and others, researchers widely rely on demonstrations to train RL policies. Therefore, for the complex tasks mentioned above, we have not yet found a RL method that can bypass the pre-train phase. Hierarchical RL may be a direction worth exploring [4].
>
> ||SAC+HER+MEGA|BC|GCPO|
> |:---:|:---:|:---:|:---:|
> |Success Rate|5.75±1.98|17.08±0.57|45.87±3.09|
>
> In summary, on complex tasks, we have not yet found an RL method that can bypass the use of demonstrations. **Although the training of GCPO depends on demonstrations, its capability to learn policies from imperfect demonstrations somewhat relaxes the conditions for using GCPO**.
>
> [1] Vinyals O, Babuschkin I, Czarnecki W M, et al. Grandmaster level in StarCraft II using multi-agent reinforcement learning[J]. nature, 2019, 575(7782): 350-354.
>
> [2] Baker B, Akkaya I, Zhokov P, et al. Video pretraining (vpt): Learning to act by watching unlabeled online videos[J]. Advances in Neural Information Processing Systems, 2022, 35: 24639-24654.
>
> [3] Ramrakhya R, Batra D, Wijmans E, et al. Pirlnav: Pretraining with imitation and rl finetuning for objectnav[C]//Proceedings of the IEEE/CVF Conference on Computer Vision and Pattern Recognition. 2023: 17896-17906.
>
> [4] Pope A P, Ide J S, Mićović D, et al. Hierarchical reinforcement learning for air-to-air combat[C]//2021 international conference on unmanned aircraft systems (ICUAS). IEEE, 2021: 275-284.
>
> > W2: Relatedly, having results for e.g. SAC + HER when doing the same thing as in appendix A.4 would be helpful.
>
> We conduct experiments related to SAC+HER with reference to the experimental setup for GCPO described in Appendix A.4. The training process curves are shown in Fig.2 in the Global Author Response PDF. It can be observed that expanding the input of the SAC+HER policy also results in a certain degree of performance degradation. We believe that the reasons are consistent with the analysis provided in Appendix A.4.

---

> > ### Comment · Reviewer_aw7R · 2024-08-12
> >
> > Thank you. I will update my score to 7 on the condition that please add these experiments and discussions to the updated manuscript.

---

> > > ### Author Response · Authors · 2024-08-13
> > >
> > > Dear Reviewer aw7R,
> > >
> > > We sincerely appreciate the time and effort you spent on our work. Your insightful comments and concerns have helped greatly in improving our paper. We will address these discussions in the final version. Thank you once again for your valuable feedback.

---

### Official Review · Reviewer_emMz · 2024-07-10

**Soundness:** 2
**Presentation:** 3
**Contribution:** 2
**Rating:** 6
**Confidence:** 3

**Summary:**

This paper proposes Goal-Conditioned Policy Optimization (GCPO), an on-policy variant for goal-conditioned RL that can also handle non-markovian reward structures. Common goal-conditioned RL methods are usually related to Hindsight Experience Replay (HER) which however can only solve tasks under the Markovian properties. Notably, GCPO leverages pre-training from demonstrations and online self-curriculum learning that can progressively select challenging goals based on the current learning process of the policy.

**Strengths:**

- well-motivated
- contributions are well outlined
- detailed ablations
- An important and interesting problem is considered

**Weaknesses:**

The paper lacks discussion of relevant works that also consider contextual/goal-conditioned RL with self-curriculum learning with similar motivations, some of which consider non-Markovian rewarded environments. Here are some of those:

- Klink et al. Self-Paced contextual reinforcement learning (CoRL 2019)
- Celik et al. Specializing Versatile Skill Libraries using Local Mixture of Experts (CoRL 2021)
- Klink et al. Self-Paced Deep Reinforcement Learning (NeurIPS 2020)
- Ottot et al. Deep Black-Box Reinforcement Learning with Movement Primitives (CoRL 2022)
- Celik et al. Acquiring Diverse Skills using Curriculum Reinforcement Learning with Mixture of Experts (ICML 2024)

please also see the questions

**Questions:**

- The method requires a desired goal distribution which might be task dependent. In which sense is this restricting the algorithm's applicability?
- How exactly is the GMM estimated? No information was given in the main text
- The work states that the self-curriculum "leads to an average 8.2% increase in policy performance" as evidenced in Table 2. However, Table 2 also shows that GCPO w/o pre-training can only achieve 4% success rate indicating that the performance boost is mainly achieved by the imitation learning policy. On the other hand, Fig.3 shows that GCPO successfully reaches more goals than the expert and BC, especially in the goal difficulty up to 0.7 but is not able to significantly reach more difficult goals. This observation is also discussed in Section 4.3.2.
While I see that the goals that are already covered by the BC can be reached successfully more often by GCPO, the self-curriculum does not seem to cover more difficult goals significantly. Section 4.3.2 suggests to "sample goals and generate demonstrations as closely as possible to the desired goal distribution". Isn't this restricting the method in the sense that the demonstration data needs to be collected accordingly?

- Connecting to the question before, is this indicating that exploring new goals that are much more difficult can not covered well by GCPO? Could this be discussed in more detail? Why is this the case? Is the KL constraint restricting the updates to more difficult goals? To my understanding, expanding to more difficult goals should be covered by the bonus (MEGA) as described in Section 3.3.

**Limitations:**

- While the detailed ablations are highly appreciated, a more thorough evaluation across more tasks is needed to assess the effectiveness of the method. This can be done by using sophisticated existing non-markovian rewarded environments, or, for demonstration purposes, adjusting the reward accordingly in existing benchmark suits such as e.g. (Meta World)

Meta World: https://github.com/Farama-Foundation/Metaworld

Please also see the weaknesses and questions.

---

> ### Author Rebuttal · Authors · 2024-08-07
>
> > Response to Q3 and Q4
>
> We consolidate your two questions into three sub-questions.
>
> 1. _The effectiveness of GCPO seems to primarily come from the pre-training, and the self-curriculum does not appear to significantly cover more difficult goals._
>
> The objective of GCRL is to achieve more goals under the expectation of $p_{dg}$, which is independent of any form of goal difficulty. Both MEGA and our GCPO do not alter the objective of GCRL, so the appropriate evaluation metric should be the success rate over $p_{dg}$. Therefore, considering the change in success rate, both pre-training and self-curriculum play significant roles. Specifically:
>
> * In GCRL, difficulty describes how easy or hard it is for a policy to achieve a goal while learning. This difficulty is closely linked to the policy's abilities, and we prefer to term it **learning-process-oriented difficulty**. Conversely, **task-oriented difficulty** pertains to the inherent challenge of a goal, unrelated to the policy's form or learning process. It is dictated by the goal itself. A goal that is task-oriented easy might be learning-process-oriented hard. There is no inherent link between these two difficulty types.
> * GCRL aims to maximize expected cumulative rewards under the goal distribution $p_{dg}$. When rewards are binary indicators of goal achievement, GCRL's objective is essentially to achieve more goals under $p_{dg}$. Hence, **GCRL's objective is inherently independent of any goal difficulty**. MEGA, while identifying goals by learning-process-defined difficulty to optimize exploration and learning, does not alter GCRL's objective. Thus, MEGA is not centered on achieving more task-defined difficult goals.
>
> * In VVC, we define $p_{dg}$ as the uniform distribution over the entire goal space. For clarity, we define task-oriented difficulty in Appendix A.1.4 and use it as the x-axis in our figures. Fig.1(a) in the Global Author Response PDF shows all discrete goals in a 3D space. Fig.1(b) depicts the goal distribution over task-oriented difficulty. It is evident that, under our definition, the distribution of desired goals over task-oriented difficulty is not uniform but rather low at both ends and high in the middle. This is exactly why our experimental results exhibits this particular shape.
>
> In essence, the valid metric for evaluating GCRL is the success rate under $p_{dg}$. Table 2 shows that the success rate of GCPO is 2.69 times that of BC, indicating that online learning with self-curriculum can significantly improve a pre-trained policy. Fig.3(b) and 3(c) are merely analyses on learned policies and learning process from the task-oriented difficulty perspective. We find that self-curriculum does gradually sample task-oriented difficult goals during training, and the resulting policies can achieve some of these goals.
>
> 2. _Does the requirement for sampling demonstrations in Section 4.3.2 limit the applicability of the method?_
>
> In Section 4.3.2, the experiment examines how demonstration goal distribution affects GCPO. Results indicate that demonstrations sampled according to $p_{dg}$ yield effective GCPO policies. We believe that this is a point to focus on for enhancing GCPO's performance, rather than a limitation of its applicability. Comparison between the results in Sections 4.3.2 and 4.2 reveals that GCPO can still train when the demonstration goal distribution differs substantially from $p_{dg}$. For instance, using $\mathcal{D}_2$ in Section 4.3.2, with 100 demonstrations (cover only 0.20% goal space) focused on difficulties within (0.2328, 0.2336), GCPO can still train a policy with a 21.86% success rate.
>
> 3. _What impact does KL have on training?_
>
> Indeed, KL impacts GCPO's training. As discussed in our response to the first sub-question, we believe that this impact is only reflected in the success rate, not in the achievement of the task-oriented difficult goals. We train GCPO with various $\lambda$:
>
> |$\lambda$|KL Value|Success Rate|
> |:-:|:-:|:-:|
> |$10^{-1}$|1.79±0.42|25.37±3.68|
> |$10^{-2}$|9.27±0.82|38.75±1.83|
> |$10^{-3}$|33.27±2.34|45.87±3.09|
> |$10^{-4}$|133.32±26.42|28.20±5.31|
> |0|368.19±15.72|25.68±4.05|
>
> As shown, a higher $\lambda$ leads to a lower KL, suggesting the GCPO policy is more statistically similar to the pre-trained policy, aligning its success rate with the pre-trained policy's 17.08%. As $\lambda$ decreases, KL increases, and the GCPO policy's success rate improves. Yet, at $\lambda < 10^{-3}$, the success rate drops, which we attribute to the weak KL constraint leading to catastrophic forgetting of the pre-trained knowledge, impeding effective policy learning.
>
> > Response to Q1
>
> The desired goal distribution is a part of the problem formulation of GCRL. We contend that this distribution, which should be a known condition, does not restrict GCPO. Even if the demonstration goal distribution significantly differs from the desired goal distribution, GCPO can still train a reasonably good policy. For a detailed analysis, please refer to our response to the second sub-problem in the first question.
>
> > Response to Q2
>
> GMM estimates a distribution with a weighted sum of multiple Gaussian kernels, $\sum_{i=1}^M \pi_i N(x|\mu_i,\Sigma_i)$, and estimate parameters with the Expectation-Maximization algorithm. Due to space constraints, we will include detailed calculation in Appendix.
>
> > Response to W1
>
> Thank you for the suggestion. We've thoroughly read the papers, with four focusing on curriculum learning and one on NMR. The key contribution of GCPO is its on-policy GCRL training framework, where self-curriculum is vital for efficient online learning. Thus, other curriculum learning methods could replace MEGA within GCPO. We will discuss these works in the related work section. The episode-based RL research offers valuable guidance for enhancing GCPO's suitability for NMR problems. We will explore this possibility in the future work.
>
> > Response to L1
>
> Please refer to Q1 in Global Author Response.

---

> > ### Comment · Reviewer_emMz · 2024-08-12
> >
> > I highly appreciate the authors' responses and additional experiments. My questions were clarified in the responses. I am adjusting the score on the condition that the authors discuss the relevant works in the final version and accordingly adjust their claims in lines 66-68.

---

> > > ### Author Response · Authors · 2024-08-13
> > >
> > > Dear Reviewer emMz,
> > >
> > > We express our heartfelt gratitude for the time and dedication you've invested in reviewing our manuscript. Your constructive feedback and perceptive comments have been immensely valuable in refining our paper. We are committed to addressing these insights in our revised work. Thank you once again for your valuable feedback.

---

### Official Review · Reviewer_U5WJ · 2024-07-13

**Soundness:** 3
**Presentation:** 4
**Contribution:** 4
**Rating:** 7
**Confidence:** 4

**Summary:**

This paper proposes a new on-policy goal conditioned reinforcement learning framework which targets non-markovian rewards, which HER based approaches are unsuccessful. GCPO is a combination of offline pre-training from expert policies using behavior cloning, and online learning from a curriculum which automatically determines complex goals outside of the agent's current goal conditioned policy. They show this frameworks success in the Velocity Vector Control task, and perform several ablation and sensitivity studies.

**Strengths:**

This paper provides a novel (as far as I am aware) framework for developing goal conditioned algorithms. This paper excels in clarity, interesting empirical and algorithmic contributions, and clear directions for future improvement.

- **Clarity:** The paper is very clearly written, and provides ample details about the algorithms and experimental design. I also appreciate the work done in the introduction and related works section, clearly grounding this work in the literature and showing the deficiencies of HER approaches.
- **Algorithmic contributions:** The GCRL framework is flexible enough to inspire a large amount of future work, and the authors do a great job enabling future research in this direction. The combination of pre-training and a goal-driven curriculum based on estimating successfully achieved goals with a GMM is interesting and novel.
- **Empirical Work:** The empirical work is well rounded, and several interesting ablations going into the specific details of all the parts of the framework is appreciated and helps readers understand further. And the success on an extremely difficult domain is a great achievement in the space.
- **Clear future directions** The authors are very clear on where their contributions are limited, and open the door to several future algorithmic contributions in this space.

**Weaknesses:**

While I think this paper would be an excellent contribution to this year's conference, there are some improvements I would like to see. I think the authors should make changes to address I-1, but the others are less consequential in my view.

- I-1. I haven't seen a discussion around how hyperparameters were tuned. While I put this work in more of a demonstration lane, but I think if the paper was clearer around how these hyperparameters were chosen, and how the baseline methods were tuned would immediately strengthen the paper. I think this is critical when discussing the results of Figure 1. Adding how extensively these methods were tuned could strengthen the evidence supporting the claims.

- I-2. While the ablation showing how the quantity of demonstration data on the final performance is appreciated, I think the lack of ablations on the quality of demonstrations is a major limitation. There are some experiments using demonstrations with different distributions of goals, these don't sufficiently capture what would happen if there is a lack of *expert* data. In many domains, there is often a lack of true expert data, but a large amount of sub-standard data. If GCPO is also performant in instances where expert data is sparse, but non-expert data isn't that would be a major achievement for the method (although my guess is issues would arise from using BC as you already state as a limitation).

**Questions:**

- How were the baselines and your method's hyperparameters tuned?

**Limitations:**

Yes.

---

> ### Author Rebuttal · Authors · 2024-08-06
>
> > Q1 & W1: How were the baselines and your method's hyperparameters tuned?
>
> We employ Grid Search to search the following hyperparameters of the evaluated algorithms:
>
> |params|search range|
> |:-|:-|
> |SAC: Network Architecture|128\*2, 128\*3, 128\*4, 128\*5|
> |SAC: ent_coef|$10^{-3},10^{-2},10^{-1}$|
> |SAC: gamma|0.99, 0.995, 0.999|
> |SAC: lr|$10^{-3},10^{-4},10^{-5}$|
> |SAC: use_sde|True, False|
> |HER: buffer_size|$10^4,10^5,10^6$|
> |HER: n_sampled_goal|2,4,8,16|
> |HER: goal_selection_strategy|episode,final,future|
> |Self-curriculum: buffer_size|$10^3,10^4,10^5$|
> |BC: Network Architecture|128\*2, 128\*3, 128\*4, 128\*5|
> |BC: l2_weight|$0,10^{-4},10^{-3},10^{-2}$|
> |BC: ent_weight|$10^{-3},10^{-2},10^{-1}$|
> |BC: batch_size|512,1024,2048,4096|
> |PPO: Network Architecture|128\*2, 128\*3, 128\*4, 128\*5|
> |PPO: ent_coef|$10^{-3},10^{-2},10^{-1}$|
> |PPO: gamma|0.99, 0.995, 0.999|
> |PPO: lr|$10^{-3},10^{-4},10^{-5}$|
> |PPO: use_sde|True, False|
>
> The other parameters are set to StableBaselines3 defaults. Taking the network architecture of BC as an example, once the optimal set of parameters has been searched, we maintain all other parameters constant and demonstrate the policy performance when using different network architectures with the following table:
>
> |Network Architecture|128\*2|128\*3|128\*4|128\*5|
> |:-:|:-:|:-:|:-:|:-:|
> |Success Rate|17.08±0.57|16.56±0.91|15.31±0.56|14.78±2.45|
>
> As [128, 128] achieved the best performance, we maintain this architecture for all experiments on BC.
>
> > W2: the lack of ablations on the quality of demonstrations is a major limitation. How does GCPO perform on non-expert demonstrations?
>
> **Firstly, we supplement experiments on demonstration quality.**
>
> In Section 4.2, $\mathcal{D}_E$ covers 10264 goals. We generate trajectories for these goals with all the policies trained in our experiments. For a specific goal, we retain only the shortest trajectory. These generated demonstrations are denoted as $\mathcal{D}'$.
>
> We use two metrics to measure demonstration quality: **Trajectory length** $I_l$. Since we employ (-1, 0) sparse rewards, this implies that shorter trajectories yield a higher cumulative reward. **Control smoothness** $I_s$. In control problems, minimal control gains is expected to reduce wear on actuators. Hence, we refer to [1] to define the control smoothness. Trajectory length and control smoothness each describe certain characteristics of demonstrations from the distinct perspectives of reinforcement learning optimization and optimal control, respectively.
>
> | Demo | #$\mathcal{D}$ | $I_l(\mathcal{D}) \downarrow$ | $I_s(\mathcal{D}) \downarrow$ |
> |:-:|:-:|:-:|:-:|
> |$\mathcal{D}_E$|10264|282.02±149.98|2.11±2.21|
> |$\mathcal{D}'$|10264|101.42±32.41|10.19±8.74|
>
> It can be observed that: From an RL perspective, $\mathcal{D}'$ is of higher quality because the trajectories are shorter, leading to a higher expected cumulative reward. From a control perspective, $\mathcal{D}_E$ is better because the trajectories are smoother.
>
> The performance of the BC policy $\pi_{BC}$ and the GCPO policy $\pi_{GCPO}$ trained on these two sets of demonstrations is:
>
> | Demo | $s(\pi_{BC}) \uparrow$ | $I_l(\pi_{BC}) \downarrow$ | $I_s(\pi_{BC}) \downarrow$ | $s(\pi_{GCPO}) \uparrow$ | $I_l(\pi_{GCPO}) \downarrow$ | $I_s(\pi_{GCPO}) \downarrow$ |
> |:-:|:-:|:-:|:-:|:-:|:-:|:-:|
> |$\mathcal{D}_E$|17.08±0.57|241.72±81.36|1.97±1.97|45.87±3.09|133.86±53.24|6.84±5.60|
> |$\mathcal{D}'$|19.10±0.22|122.81±54.36|8.85±9.60|39.26±2.02|150.59±63.89|18.11±12.69|
>
> The results reveal that $\pi_{BC}$ closely aligns with the demonstrations on both quality metrics, indicating that demonstration quality has a direct impact on BC. Additionally, the BC policy trained on $\mathcal{D}'$ has a slightly higher success rate, which we speculate is due to $\mathcal{D}'$ being more suitable for RL (the network architecture and training hyperparameters used to generate $\mathcal{D}'$ are the same as those for the BC policy). However, after the self-curriculum learning, the GCPO policy corresponding to $\mathcal{D}_E$ performs better and exhibits a shorter trajectory length. This suggests that the influence of demonstration quality on GCPO's online learning may not be as direct as pre-training, and further research is required to understand this relationship.
>
> In summary, on one hand, it is challenging to define demonstration quality suitable for RL through a few metrics [2]. This is a research direction that deserves further exploration. On the other hand, demonstration quality does affect GCPO pre-training. How demonstration quality potentially influences the online self-curriculum learning of GCPO remains an intriguing question for further exploration.
>
> **Secondly, GCPO is capable of training well-performed policies from non-expert demonstrations.**
> Please refer to the response to Q2 in the Global Author Response for a detailed analysis.
>
> [1] Mysore S, Mabsout B, Mancuso R, et al. Regularizing action policies for smooth control with reinforcement learning[C]//International Conference on Robotics and Automation. 2021.
>
> [2] Belkhale S, Cui Y, Sadigh D. Data quality in imitation learning[C]//Advances in Neural Information Processing Systems. 2024.

---

> ### Author Response · Authors · 2024-08-13
>
> Dear Reviewer U5WJ, Do you have any further concerns? Please let us know, we will try our best to address them quickly. We sincerely anticipate your response.

---

> > ### Comment · Reviewer_U5WJ · 2024-08-13
> >
> > Sorry, this paper got missed when I was responding to author comments. you have addressed the small concerns I had for this paper. I especially appreciate the extra results on demonstration quality and performance, I think this is particularly important whenever we are using data collected using unknown policies.

---

> > > ### Author Response · Authors · 2024-08-14
> > >
> > > Dear Reviewer U5WJ,
> > >
> > > We sincerely appreciate the time and effort you spent on our work. We will address the above discussions in the final version. Thank you once again for your valuable feedback.

---

### Author Rebuttal · Authors · 2024-08-07

We sincerely appreciate your valuable feedback and careful review of our paper. We address the two common concerns of many reviewers in the following:

> Q1: How does GCPO perform on other RL domain tasks?

**We conduct two sets of experiments to demonstrate the general applicability of our method.**

**Environments**: For the first set of experiments, we conduct evaluations on a customized PointMaze environment (PointMaze_Large_DIVERSE_G-v3) from Gymnasium-Robotics [1] within the Mujoco physics engine. The only modification we made to the environment is to expand the number of desired goals from 7 to 45, making our customized version of PointMaze more challenging than the original version. For the second set of experiments, we employ a customized Reach (PandaReach-v3) task on the Franka Emika Panda robot physics engine. The Reach task is akin to those in the Meta-World environment, both involving robotic arm tasks where the objective is to reach a specified goal state. The only modification we made to the environment is to change the distance_threshold used to determine goal reaching from 0.05 to 0.01. Consequently, our customized version of the Reach task has a stricter criterion for determining goal arrival, making it more difficult than the original version of Reach.

**Reward Settings**: The original rewards for both the PointMaze and Reach tasks are Markovian. To evaluate the performance of our algorithm under different NMR settings, we design two distinct types of NMRs. For the PointMaze, the NMR we designed is: the task is considered successful only if, after the point reaches the goal, it moves away by at least a certain distance and then returns to the goal. For the Reach task, the NMR we designed is: the Panda robot must first pass through a specific waypoint before reaching the goal to be considered successful, and each goal has a different waypoint. Both of these settings strictly adhere to the definition of NMR, where the reward is defined by the states and actions over multiple steps.

**Demonstrations**: The demonstrations for PointMaze are sourced from Minari [3] (pointmaze-large-v1), while the demonstrations for Reach are generated by us, with reference to the PID controller as described in the official documentation [4].

**Performance**: We evaluate SAC+HER+MEGA, BC, and GCPO on the PointMaze and Reach tasks under both MR and NMR settings. The following table presents the success rates of these algorithms. It can be observed that under the MR settings, GCPO exhibits similar performance to SAC+HER+MEGA. However, under the NMR settings, where HER cannot be effective, the performance of GCPO is significantly better than that of SAC. Taking into account the performance of GCPO on the VVC task as illustrated in the main paper, we showcase the general applicability of GCPO across a variety of tasks.

|Task|Reward|SAC+HER+MEGA|BC|GCPO|
|:---:|:---:|:---:|:---:|:---:|
|Reach|MR|100.0±0.0|70.63±2.99|100.0±0.0|
|Reach|NMR|0.72±1.34|10.52±11.70|80.26±17.01|
|PointMaze|MR|100.0±0.0|75.96±5.34|93.33±3.06|
|PointMaze|NMR|4.17±0.93|22.8±3.71|47.50±8.06|

_**Note**: Due to the inability of HER to get rewards in the NMR settings, in the experiments, SAC+HER+MEGA is employed for the MR settings, while SAC is employed for the NMR settings._

[1] https://robotics.farama.org/

[2] Gallouédec Q, Cazin N, Dellandréa E, et al. panda-gym: Open-source goal-conditioned environments for robotic learning[C]//4th Robot Learning Workshop: Self-Supervised and Lifelong Learning@ NeurIPS 2021. 2021.

[3] https://minari.farama.org/

[4] https://panda-gym.readthedocs.io/en/latest/usage/manual_control.html

> Q2: How does GCPO perform on non-expert demonstrations?

**GCPO is capable of training well-performed policies from non-expert demonstrations.** The intrinsic reason is that GCPO employs online learning to fine-tune pre-trained policies. Consequently, even if the demonstrations are non-expert and the pre-trained policies perform poorly, GCPO can still continuously optimize these policies through online learning.

In Section 4.2, although the average trajectory length of $\mathcal{D}_E^0$ reached 281.83, covering only 20.24% of goal space, the GCPO policy trained on it achieves a success rate of 45.87%, with an average trajectory length of 134.47. This comparison indicates that $\mathcal{D}_E^0$ consists of non-expert demonstrations. On the other hand, in contrast to $\mathcal{D}_E^3$, which covers 78.55% of the goal space with an average trajectory length of 116.56, $\mathcal{D}_E^0$ is only a quarter in size and has trajectories that are 2.42 times longer, implying a substantial decrease in its quantity and quality. Nonetheless, the GCPO policy trained on $\mathcal{D}_E^0$ achieves 76.58% of the success rate of the policy trained on $\mathcal{D}_E^3$.

Therefore, we contend that GCPO has the capability to train from non-expert demonstrations.

---

### Author Response · Authors · 2024-08-12

Dear Reviewers: We are approaching the conclusion of the author-reviewer discussion period, which is set to end in one-or-two days. We kindly request that you review the rebuttal. Should you have any further questions or require clarification on any points, please feel free to reach out. We are committed to addressing your queries promptly. We greatly value your feedback and look forward to your insights.

---

### Decision · Program_Chairs · 2024-09-25

**Decision:**

Accept (poster)

**Comment:**

The paper presents an on-policy Goal-Conditioned Reinforcement Learning (GCRL) framework (GCPO), which addresses the limitations of existing HER-based methods, in handling non-Markovian rewards (NMR). GCPO combines pre-training from demonstrations with an online self-curriculum learning strategy, allowing it to progressively tackle more challenging goals based on the agent's current capabilities. The authors demonstrate the efficacy of GCPO on a challenging task involving fixed-wing UAV velocity vector control, showcasing its applicability to both Markovian and non-Markovian reward problems. The paper shows the strength in handling both Markovian and non Markovian reward scenarios. The rebuttal helped clarify some concerns such as that on general applicability to diverse tasks and demonstration quality. Given the overall contributions and its strong empirical performance, I recommend the paper for acceptance with a revision to address concerns raised through the reviewer discussions.